# Profiling water vapor mixing ratios in Finland by means of a Raman lidar, a satellite and a model

Maria Filioglou[1], Anna Nikandrova[2], Sami Niemelä[3], Holger Baars[4], Tero Mielonen[1], Ari Leskinen[1, 5], David Brus[3], Sami Romakkaniemi[1], Elina Giannakaki[1, 6] and Mika Komppula[1]

[1] Finnish Meteorological Institute, Atmospheric Research Centre of Eastern Finland, Kuopio, Finland
[2] Department of Physics, University of Helsinki, Helsinki, Finland
[3] Finnish Meteorological Institute, Helsinki, Finland
[4] Leibniz Institute for Tropospheric Research (TROPOS), Leipzig, Germany
[5] Department of Applied Physics, University of Eastern Finland, Kuopio, Finland
[6] Department of Environmental Physics and Meteorology, University of Athens, Athens, Greece

*Correspondence to*: Maria Filioglou (maria.filioglou@fmi.fi)

**Abstract.** We present tropospheric water vapor profiles measured with a Raman lidar during three field campaigns held in Finland. Co-located radio soundings are available throughout the period for the calibration of the lidar signals. We investigate the possibility of calibrating the lidar water vapor profiles in the absence of co-existing on-site soundings using water vapor profiles from the combined Advanced InfraRed Sounder (AIRS) and the Advanced Microwave Sounding Unit (AMSU) satellite product; the Aire Limitee Adaptation dynamique Development INternational and High Resolution Limited Area Model (ALADIN/HIRLAM) Numerical Weather Prediction (NWP) system, and the nearest radio sounding station located 100 km away from the lidar site (only for the permanent location of the lidar). The uncertainties of the calibration factor derived from the soundings, the satellite and the model data are < 2.8 %, 7.4 % and 3.9 %, respectively. We also include water vapor mixing ratio intercomparisons between the radio soundings and the various instruments/model for the period of the campaigns. A good agreement is observed for all comparisons with relative errors that do not exceed 50 % up to 8 km altitude in most cases. A four-year seasonal analysis of vertical water vapor is also presented for the Kuopio site in Finland. During winter months, the air in Kuopio is dry ($1.15 \pm 0.40$ g kg$^{-1}$); during summer it is wet ($5.54 \pm 1.02$ g kg$^{-1}$); and at other times, the air is in an intermediate state. These are averaged values over the lowest 2 km in the atmosphere. Above that height a quick decrease of water vapor mixing ratios is observed, except during summer months where favorable atmospheric conditions enable higher mixing ratio values at higher altitudes. Lastly, the seasonal change in disagreement between the lidar and the model has been studied. The analysis showed that, on average, the model underestimates water vapor mixing ratios at high altitudes during spring and summer.

## 1 Introduction

The radiative balance between incoming solar radiation and outgoing longwave radiation is the primary regulator of Earth's climate. Changes in atmospheric components, such as aerosols and greenhouse gases which affect the radiative balance, have an impact on climate (McCormic and Ludwig, 1967; Twomey, 1974; Lohmann and Feichter, 2005; Boucher et al., 2013). As

the dominant greenhouse gas, water vapor is considered to be the main feedback agent of the atmosphere's state (Held and Soden, 2000; Dessler et al., 2008). As its concentration mostly depends on the air temperature, climate models suggest an amplified initial warming effect in global warming scenarios (IPCC, 2013). This important feedback roughly doubles the amount of warming caused by carbon dioxide (Held and Soden, 2000; Soden et al. 2002; Soden et al., 2005). In addition it is

also involved in most of the atmospheric processes, such as frontal generation systems (Van Baelen et al., 2011), cloud formation, atmospheric mixing, photochemical processes (McCormack et al., 2008) and aerosol hydration (Feingold et al., 2003; Estillore et al., 2016).

The validation of numerical weather forecast and climate models usually falls to the low spatial and temporal resolution of the observational parameters. Unlike other greenhouse gases, water vapor, can be highly variable both in space and time,

making it hard to simulate. To help address this issue more accurate and nested observational data are needed. Retrievals from space-borne passive sensors can provide some information but their vertical resolution is insufficient given the frequent occurrence of strong vertical gradients.

Based on the measuring technique, water vapor mixing ratios (WVMR) can be separated into two major categories. The first category considers in-situ measurements of temperature and relative humidity which can be converted into WVMR. A

plethora of weather stations provide ground based WVMR over the globe. Nevertheless, such measurements are not representative for the whole atmosphere since their spatial availability is poor over remote areas (e.g. over oceans) and no vertical information is provided. Vertical mixing ratio profiles with high accuracy are delivered by means of radiosondes. Radiosondes (RS) are a common and reliable in-situ technique but they also lack temporal and spatial coverage as the number of sites is rather low and most of them perform very few soundings per day. Furthermore, the wind-driven drifting of

the device can be misleading in terms of geographical location of the vertical information. In contrast, remote sensing techniques such as microwave radiometers (England et al., 1992; Reagan et al., 1995), differential absorption lidars (DIAL) (Bösenberg, 1998), photometers (Barreto et al., 2013) and Raman lidars (Ferrare et al., 1995; Turner et al., 2001; Whiteman, 2003; Leblanc et al., 2012; Navas-Guzmán et al., 2014; Foth et al., 2015) have been successfully adopted in water vapor studies. While microwave radiometers and photometers can accurately deliver the total precipitable water vapor (TPW),

lidars (DIAL and Raman) are the only instruments available for high temporal and vertical resolution of continuous WVMR measurements. DIAL lidars are able to provide accurate high-resolution profiles of water vapor but their complex laser transmitter setup make their WVMR automatization a difficult task. Raman lidars use a simpler setup than DIAL lidars, although the majority of them are limited to nighttime performance due to collection of the intense daytime background light by the weak Raman-shifted channels. Nonetheless, Foth et al. (2017) proposed a methodology to retrieve water vapor mixing

ratios during daytime by using a microwave radiometer and the Raman lidar profiles.

In the present paper, we calibrate Raman lidar WVMR profiles using in-situ, satellite and model data. We have used RS, retrievals from AIRS/AMSU instruments on board of Aqua satellite (Parkinson, 2003) and modeled WVMR from ALADIN/HIRLAM NWP model (Seity et al., 2011; Bengtsson et al., 2017). Our dataset concentrates on three field campaigns conducted during 2014 and 2015. Within this time frame, 723 radio soundings were performed in total, but not all

of them were suitable for direct comparison with the lidar due to the daytime limitation or occurrence of low-level clouds. Furthermore, we derive the seasonal variation of WVMR at Kuopio site where the lidar instrument is permanently located, both from the lidar and model, thereby validating the accuracy of the model and the capabilities of the lidar under the demanding low-water content conditions of Finland.

The outline of the paper is as follows. First, we give a short description of the instruments/model (Sect. 2) followed by the methodology used to calculate/extract the WVMR profiles from the various sources (Sect. 3). An overview of the existing calibration methods for the lidar is also given in Sect. 3. The calibration factors from the various reference instruments are calculated in Sect. 4. Section 4 also includes comparisons between the RS, and the various instruments and the model, under cloud-free conditions. Furthermore, Sect. 5 presents the seasonal variability of the WVMR at the Kuopio site in Finland and

the seasonal discrepancies between the lidar and the model based on a four-year period of lidar and model data. Our summary and concluding remarks are given in Sect. 6.

## 2 Instrumentation

The site locations where the three field campaigns took place cover Finland from the southwest to the north: Hyytiälä (61.84$^o$ N, 24.29$^o$ E, 179 m a.s.l) from 1$^{st}$ of April to 29$^{th}$ of September 2014; Kuopio (62.73$^o$ N, 27.54$^o$ E, 190 m a.s.l) from

11$^{th}$ to 29$^{th}$ of May 2015; and Pallas (67.99$^o$ N, 24.24$^o$ E, 345 m a.s.l) from 22$^{nd}$ of September to 5$^{th}$ of December 2015. Further information on the site locations and available instrumentation can be found in Hatakka et al. (2003) and Hirsikko et al. (2014).

### 2.1 Remote sensing data

### 2.1.1 The Polly$^{XT}$- FMI

The lidar data were obtained with the fully automated and portable multi-wavelength Raman lidar Polly$^{XT}$ (Althausen et al., 2009; Engelmann et al., 2016) operated by the Finnish Meteorological Institute (FMI). The system is a Raman polarization lidar with water vapor capabilities. The detection is performed at the three emitted elastic wavelengths (355 nm, 532 nm and 1064 nm) and the three inelastic Raman-shifted wavelengths (387 nm, 407 nm and 607 nm). Information on the polarization of the signal is available at 532 nm – cross polarization with respect to the initial emitted polarization plane. The biaxial

system attains full overlap at 800-900 m (Engelmann et al., 2016). Below that height, signals are corrected with an overlap function introduced in Wandinger and Ansmann (2002). The instrument operates with a spatial resolution of 30 m and a temporal resolution of 30 s. Near real-time measurements are visualized through the lidar network PollyNET (Althausen et al., 2013; Baars et al., 2016) and can be accessed through the following webpage: http://polly.tropos.de/.

### 2.1.2 Satellite data – AIRS/AMSU

Both AIRS (Aumann et al., 2003) and AMSU (Revilla et al., 1997) are instruments on-board the Aqua satellite (Parkinson, 2003). AIRS is a thermal IR grating spectrometer which allows measurements of temperature and humidity as a function of altitude. It has 2378 detectors in four wavelength bands: 3.74 μm, 4.61 μm, 6.20 to 8.22 μm, and 8.80 to 15.40 μm. AMSU is
a 15-channel microwave sounder providing temperature and humidity information along the track. In this study, we used the combined level two (L2) version 6 support products (AIRS + AMSU) (AIRS Science Team, 2013) provided publicly by NASA ([http://disc.sci.gsfc.nasa.gov/uui/datasets?keywords=%22AIRS%22](http://disc.sci.gsfc.nasa.gov/uui/datasets?keywords=%22AIRS%22)) to surpass the limitation of AIRS and deliver usable water vapor profiles under cloudy conditions. Hearty et al. (2014) report on instrumental biases of AIRS/AMSU concluding to up to 2 K for the temperature measurements and 10 % wet for the water vapor, where the bias is largest within
the boundary layer. A detailed description of the uncertainties in the retrievals can be found in Hearty et al. (2014) and the AIRS version 6 performance and test report. The combined AIRS/AMSU product is reported in a 50 km spatial resolution at nadir and covers 100 vertical pressure levels.

### 2.2 In situ data – Radiosondes

During the intensive campaign periods 630, 66 and 27 radio soundings were performed at the Hyytiälä, Kuopio and Pallas
sites, respectively. For the Hyytiälä campaign soundings were performed four times per day at 5:20, 11:20, 17:20 and 23:20 UTC. During the Kuopio campaign, RS were performed three times per day at 13:00, 19:30 and 22:00 UTC, increasing their launching frequency during the last two days of the campaign. Lastly, during the Pallas campaign the soundings were performed less frequently and in a non-standardized way in terms of time, focusing on special events. In the first two campaigns the RS was launched from a location not more than 100 m away from the lidar whereas in Pallas the
launching site was 5 km away from the lidar site. The radiosondes used are of Vaisala type RS41 (Kuopio) and RS92 (Hyytiälä and Pallas). RS data from Jyväskylä airport (RS92), the nearest RS station located about 100 km away from Kuopio, were also used in this study. For temperature and relative humidity measurements, the RS41 has associated **instrumental** uncertainties of $0.3\,^{\circ}C$ and 4 % in the first 16 km in the atmosphere, respectively. According to the manufacturer's specifications, the difference between RS41 and RS91 are within $0.1\,^{\circ}C$ and 2 % for the same height range.

### 2.3 Model data – ALADIN/HIRLAM NWP system

The ALADIN-HIRLAM cooperation is an international effort of 26 countries (mainly from Europe) to develop a mesoscale weather forecasting system. One configuration of the common ALADIN/HIRLAM NWP system, HARMONIE/AROME (Bengtsson et al., 2017), has been used operationally at FMI since 2006. HARMONIE/AROME is a non-hydrostatic model based on fully compressible Euler equations, where the time integration of the equation set is handled with two time level
semi-implicit semi-Lagrangian advection scheme. The model's physical parameterization package includes treatment of sub-

grid scale processes related to cloud microphysics, turbulence, radiation transfer, shallow convection, surface and soil. All the parameterization schemes are described in detail by Bengtsson et al. (2017) and Seity et al. (2011).

In this study we have used data from FMI's operational HARMONIE/AROME setup. During the study period, two development versions of the model have been in use: i) cy38h1.1 (Jan. 2014 – Mar. 2015, Niemelä, 2015) and ii) cy38h1.2 (Mar. 2015 onwards). The main difference between these two versions is related to cloud processes, where the fast liquid water process is rigorously separated from slower ice water process in cy38h1.2 (Ivarsson, 2015). The new model version has maximum WVMR bias up to $< 0.12$ g kg$^{-1}$ compared to the older version where biases up to 0.20 g kg$^{-1}$ were observed in the first 4 km in the atmosphere. The horizontal grid size is 2.5 km $\times$ 2.5 km with 65 levels in vertical. In both versions, 49 vertical levels are located within the lowest 8 km. The model is initialized every 3 hours by using 3-dimensional variational algorithm (3D-Var) with conventional observations from TEMP (upper air soundings), SYNOP (surface synoptic observations), AMDAR (aircraft meteorological data relay), SHIP (ship synoptic code) and DRIBU (drifting buoys).

The hourly profiles of specific humidity, temperature and pressure were extracted from the model data for the locations: Hyytiälä, Kuopio and Pallas sites. The dataset included short forecasts (+3h…+8h) from subsequent model runs initiated at 00:00, 06:00, 12:00 and 18:00 UTC. The data was interpolated bi-linearly in the horizontal, whereas full resolution was used in the vertical.

## 3 Methodology

The WVMR is defined as the ratio of the mass of water vapor to the mass of dry air in a given volume can be calculated as:

$$w(r) = \frac{MW_{H_2O}}{MW_{DryAir}} \frac{N_{H_2O}(r)}{N_{DryAir}(r)} \sim 0.78 \frac{MW_{H_2O}}{MW_{DryAir}} \frac{N_{H_2O}(r)}{N_{N_2}(r)}, \tag{1}$$

where $MW_{DryAir}$ and $MW_{N_2}$ are the molecular weights of water vapor and nitrogen. $N_{H_2O}(r)$, $N_{N_2}(r)$ and $N_{DryAir}(r)$ denote the molecular number densities of the two atmospheric gases and dry air at altitude r. The 0.78 value stands for the fractional volume of nitrogen in the atmosphere. The latter expression is utilized in the water vapor Raman lidar technique, and is proportional to the water vapor mixing ratio.

### 3.1 WVMR profiles from Raman lidar signals

The Raman lidar WVMR technique has been extensively discussed in literature (e.g. Whiteman, 2003). The approach is based on collecting the vibrational Raman backscattered signals from water vapor at 407 nm and nitrogen molecules at 387 nm, both excited from 355 nm wavelength light. We calculate the WVMR from the lidar signals as:

$$w(r) = K \frac{P(r, \lambda_{H_2O})}{P(r, \lambda_{N_2})} \exp\left(\int_0^r [\alpha(r', \lambda_{H_2O}) - \alpha(r', \lambda_{N_2})] dr'\right), \tag{2}$$

where $P(r, \lambda_{N_2/H_2O})$ is the measured range-dependent backscatter signal, $K$ is the lidar system calibration factor and $\alpha(r', \lambda_{N_2/H_2O})$ the extinction coefficients caused by the two gases. The exponential component accounts for the different

attenuation of the returned signal with $\alpha(r', \lambda_{N_2/H_2O})$, including both the molecular and the particle contribution. In this paper the particle extinction contribution is neglected as the resulting error is less than 1.3 % at 2 km altitude as calculated by Foth et al. (2015). Whiteman (2003) concluded that such assumption can introduce an error of approximately 2 % for every 0.5 of AOD change below 2 km altitude. The lidar system calibration factor ($K$) includes the range-independent lidar constants for the two Raman-shifted wavelengths, the Raman backscatter cross sections of $N_2$ and $H_2O$, the ratio of the molecular masses and the fractional volume of nitrogen (0.78). For the calculation of the molecular density a-priori knowledge of vertical profiles of temperature and pressure are needed. We have calculated all lidar-derived WVMR using information provided by the radiosondes. The use of different input data, such as from the satellite or the model can introduce $< 0.10$ % and $< 0.32$ % maximum averaged discrepancies valid for the whole atmospheric column up to 8 km compared to that of the RS, respectively.

## 3.2 Lidar water vapor calibration methods

The calibration factor can be derived in two different ways. The first requires precise knowledge of the ratio of the lidar channel transmission coefficients and the Raman cross sections for the two active channels. Previous studies using this approach (Vaughan et al., 1988; Leblanc et al., 2012) computed the calibration constant with a 10-12 % accuracy. The second approach determines the calibration factor using simultaneous measurements from a collocated reference instrument. Approaches on this second technique include water vapor comparisons with radiosondes, satellites and microwave radiometers (Ferrare et al., 1995; Mattis et al., 2002; Miloshevich et al., 2004; Madonna et al., 2011; Leblanc et al., 2012; Reichardt et al., 2012; Navas-Guzmán et al., 2014; Foth et al., 2015). The accuracy of the calibration factor derived using these techniques fluctuates between 5 and 10 %. Since it is rather challenging to decrease the uncertainties in the Raman cross section calculations and define the optical transmission characteristics, we adopted the second approach.

There are several methods to calculate the calibration factor with this second approach. The principle of the first method is to perform a linear regression between the uncalibrated WVMR lidar signal and the known WVMR from the RS or any other reference instrument (England et al., 1992). The calculated slope is the unknown calibration factor (regression method). In general, a set of such comparisons is performed to increase the statistical significance of the derived factor. However, small changes in the lidar set up such as change of neutral density filters requires the calculation of a new factor. A second method falling into the same calibration category is to take into account the simultaneous total precipitate water (TPW) from a microwave radiometer (Madonna et al., 2011; Foth et al., 2015) or any other instrument capable of delivering an equivalent information. By integrating the lidar's WVMR profiles, the two quantities become comparable and it is then possible to compute the calibration factor. Lastly, the profile method (Reichardt et al., 2012) estimates the calibration factor by matching the mixing ratio profiles from the lidar and the reference instrument in a certain height range. The factor fulfilling this requirement is the optimum one. In this study we used the regression method since it is the method introducing the best accuracy and can be applicable in most days assuming that the calibration area is cloud-free.

## 4 Lidar calibration

For the calibration of the lidar signal cloud-free nighttime atmosphere is considered. The lidar data were averaged over 30 mins centered on the RS launch time. The RS vertical resolution was interpolated to the lidar's grid, where a 90 m unweighted sliding-average was applied to smooth the signal. Only lidar signals between 0.5 and 3.5 km were retained for the calibration. This height limitation was used to minimize the inaccuracies in the WVMR values due to possibly different overlaps between the channels used for the water vapor calculation (see Sec. 4.1) and major drifts in the RS at higher altitudes. In unstable atmospheric conditions these two could result measuring very different atmospheric layers showing temporal and spatial mismatching (Brocard et al., 2013) (see also Sect. 4.1). However, the upper height limit was lowered from 3.5 km when lidar signal-to-noise ratio (SNR) values were lower than two. For the satellite data we performed a trajectory analysis, forward or backward depending on the overpass and the RS launch time, for each of the satellite footprints selecting the profile whose trajectory endpoint was closest to the site. This method was used because it produced better agreement between the satellite data and the radio soundings than the usage of the closest satellite pixel to the site. The trajectory analysis was performed using the HYSPLIT model (HYbrid Single Particle Lagrangian Integrated Trajectory, Draxler and Hess, 1998; Stein et al. 2015). Satellite data marked with quality assurance flag 2 are omitted (AIRS version 6, level 2, Kahn et al., 2012) while the maximum overpass difference between the on-site RS and the satellite was set to six hours. For NWP model data the profile closest in time was used. In each campaign model's grid box was centered on the lidar site. A fixed intercept of 0 was used in the regression analysis in order not to introduce positive WVMR at heights where none should exist.

An example case (23[rd] of May 2015) demonstrating the calibration method with all the available reference instruments and the model is shown in Fig. 1. The satellite overpass is from 00:28 UTC on 24[th] of May 2015. The closest satellite footprint was 14 km away from the site (white dot) and the selected WVMR for this case was based on the trajectory analysis which originated at the green dot shown in Fig. 1a. Figure 1b shows the range corrected signals at 1064 nm along with the RS launch times marked as vertical red lines. The RS was launched at 22:00 UTC, and averaged lidar signals between 21:45 and 22:15 UTC were used (white rectangle). Applying the slope method, we obtained calibration factors of $17.87 \pm 0.17$ g kg$^{-1}$, $17.43 \pm 0.13$ g kg$^{-1}$ and $17.44 \pm 0.16$ g kg$^{-1}$ for RS, AIRS/AMSU and ALADIN/HIRLAM, respectively (Fig. 1c). The uncertainty reported here indicates the standard error of the slope.

From the 630 soundings in Hyytiälä, 66 in Kuopio and 27 in Pallas, 23, 10 and 5 RS are suitable for the calibration procedure based on background light and cloud conditions. Due to the high geographical latitude and the time of the campaigns in Hyytiälä and Kuopio (between May and September), Finland's background sky light is too intense for this technique to retrieve water vapor profiles from lidar observations. For example, over the 18-day campaign in Kuopio, daytime increased by two hours. For all suitable aforementioned calibration cases factors from the various instruments/model were calculated and are summarized in Table 1 for Hyytiälä and Pallas and in Table 2 for Kuopio. The overall calibration

factor computed has an associated uncertainty of <1 % for the RS launched on site and 2.8 % for the nearest RS (only for Kuopio) corresponding to a mean factor of $17.46 \pm 0.13$ g kg$^{-1}$ and $16.94 \pm 0.48$ g kg$^{-1}$, respectively. The uncertainty for the satellite and the model fluctuated at 7.4 % and 3.9 % with a mean factor of $18.53 \pm 1.37$ g kg$^{-1}$ and $17.78 \pm 0.69$ g kg$^{-1}$, respectively. These accuracies comply with previous observational studies (Ferrare et al., 1995; Navas-Guzmán et al., 201l)

yet we are aware of none publications regarding the calibration of lidar WVMR signals with the use of a model. We observe that the satellite-derived calibration factor diverges from the RS-derived by about 6 %. This deviation is interpreted, depending on the water vapor amount, to as high as 0.4 g kg$^{-1}$ offset for mixing ratios of about 8 g kg$^{-1}$ and 0.1 g kg$^{-1}$ offset for drier conditions when calibrating with the satellite. Such bias can have a large impact, for example in changing environments such as the tropopause, where the radiative forcing of surface climate is being calculated (Leblanc et al., 2012;

Müller et al., 2016).

It can be seen in Tables 1 and 2, that there are a couple of cases where the satellite or/and the model have underestimated/overestimated the mixing ratios, hence the calibration factor. Consequently, individual cases can have a large effect on the calculated calibration factor thus, a more robust factor was retrieved by applying the regression method to all cases together (Fig. 2). Discrepancies are much lower when using this method as all techniques converge to the value

calculated with the on-site RS, which is presumably the closest one to the true factor. If available, we suggest using this technique when calibrating lidar WVMR signals. We should mention here that the calculation of an overall factor is possible since in all three campaigns the lidar set up had the same configuration for the channels relating to water vapor.

## 4.1 Intercomparisons

A statistical analysis between on-site RS and the rest (lidar, satellite, model and nearest RS) is presented here by calculating

absolute and relative differences from the on-site RS (Figs. 3a-h). Cases during daytime are excluded when RS profiles are compared with the lidar. For the satellite comparisons, a suitable WVMR profile was selected based on the trajectory analysis. For the ALADIN/HIRLAM comparisons, the profile from the time point closest to the RS launch time was selected. All cases presented here are cloud-cleared. Furthermore, the RS from Jyväskylä airport at 18:00 UTC is compared with the on-site RS during the Kuopio campaign to evaluate the nearest available RS. For the lidar data, a 90 m, 270 m and

390 m vertical smoothing is considered up to 3 km, 3 to 5 km and above 5 km, respectively while signals with SNR less than two are discarded.

Absolute deviations between the RS and Polly$^{XT}$ observations are below 0.2 g kg$^{-1}$ at altitudes above 0.5 km (Fig. 3a). The largest absolute discrepancies are observed in the lowermost part of the atmosphere between the surface and 0.5 km. Although in relative error terms these discrepancies are not of major importance, they show possible instrumental limitations

which result from the optical alignment region of the water vapor related channels which are focussing on the far range. For the seasonal analysis (see Sect. 5), we assumed well mixed conditions for the first 0.5 km keeping a constant value down to surface. We have also found that there is a better agreement between the lidar and the RS in the first 4 km. Above that the

relative error is bigger which is mainly attributed to a combination of low water vapor content and drifting of the RS device (Brocard et al., 2013). In all cases the relative error stayed well below 35 % (Fig. 3b).

The comparison between RS and satellite observations is shown in Figs. 3c and 3d. While absolute deviations are well below 0.85 g kg$^{-1}$, relative fluctuations of up to 200 % were observed. These large relative values are most likely caused by the larger spatial resolution in satellite data compared to that of RS measurements. On top of that, the sparser vertical resolution in satellite data cannot accurately attribute the geometrical boundaries of the layers as seen by the lidar. Accounting for these two factors, in the presence of strong vertical gradients one should expect WVMR inconsistencies between these two instruments. Mamouri et al. (2007) found that the differences between the lidar and the satellite are larger between 1.0 and 5.0 km, a feature which we have also observed. This behaviour is most likely the effect of the geometrical boundaries of the water vapor layers which cannot be precisely defined in the satellite. We should note here that the trajectory analysis for the overpass selection introduces smaller deviations when comparing with the RS and/or the lidar, and should be preferred over the closest overpass when there is sufficient time difference. However, since orographic lifts can modify air mass properties it should be used with care.

The relative difference between RS and the model is less than 36 % for the lowest 2 km (Figs. 3e-f). Above 2.5 km, deviations of up to 55 % were found. There is a constant positive bias above 5 km between the model and RS during the Hyytiälä campaign, which is not present in the other two campaigns, indicating a possible model-version dependence as an older model version is used for the Hyytiala campaign. Averaged specific humidity biases over Scandinavia showed a dry bias in the older version which has been reduced in the new one and could be the source of this behaviour.

Lastly, the nearest available radio soundings from Jyväskylä and the RS launched at Kuopio site are compared in Figs. 3g and 3h. The data considers only the Kuopio campaign since the Polly$^{XT}$ is permanently located there. On-site soundings at 13:00, 19:30 and 22:00 UTC are compared with the 18:00 UTC RS launched in Jyväskylä. The discrepancies between the local RS and the on-site are smaller in the evening or at night compared with those at 13:00 UTC. Most probably the mesoscale meteorology along with planetary boundary layer (PBL) growth/collapse times resulted in these discrepancies and should be taken into consideration when using distant RS for further use (e.g. in WV lidar calibration or calculation of molecular coefficients in lidar retrievals).

# 5 Seasonal variation of water vapor mixing ratio in Kuopio

## 5.1 Seasonal water vapor from Polly$^{XT}$ lidar data

Since 16$^{th}$ of November 2012, the Polly$^{XT}$ lidar has been located at Kuopio site operating automatically 24/7. For the seasonal analysis, measurements from a four-year period between November 2012 and August 2016 were selected (campaigns outside of Kuopio were excluded). One WVMR profile was calculated per day, preferably during the darkest hour. For the Kuopio region this is around 22:30 UTC throughout the year. Profiles were averaged over 30 minutes in time and the applied vertical smoothing is the same as that described in Sect. 4.1. As indicated in Sect. 4.1, lidar profiles from 0.5

km down to the surface were kept constant based on the WVMR value at 0.5 km. Furthermore, when necessary the lidar was calibrated using the closest RS from the Jyväskylä site. Figure 4 shows the number of available monthly nighttime profiles for the chosen period and their percentage share in each month. A total of 388 available measurements indicated with the green color in Fig. 4 were further analyzed to extract the seasonal variability. Vertical mean profiles were calculated for each month while the seasonal value was retrieved by averaging the corresponding three months. The results are shown in Figs. 5a to 5d.

In Table 3, seasonal means were calculated for 2 km segments of atmosphere up to 8 km. During summertime (JJA), the highest concentration of WV is observed with a mean value of $5.54 \pm 1.02$ g kg$^{-1}$ for the lowest 2 km. While September's WV mixing ratios are comparable to those of July and August, the mean value for autumn (SON) decreases rapidly up to the driest period in winter (DJF), with $1.15 \pm 0.40$ g kg$^{-1}$ for the first 2 km and rapid drying above that. This follows the annual trend in temperature where the lowest values are observed at that period, especially during January, indicating absence of moisture. Spring and autumn have about the same mean amount of WV at all attitudes although individual months show a variety of mixing ratios.

## 5.2 Discrepancies between the lidar and the model

Following identical procedures to those described in Sect. 5.1, we derived the seasonal variability of WV using ALADIN/HIRLAM data including only the lidar-suitable dataset used in the previous section. In Fig. 6a we present seasonal mean WVMR between the lidar and the model-limited dataset. Fig. 6b and Fig. 6c show the absolute and relative biases. Both the lidar and the model are in reasonable agreement, although small discrepancies can be observed. At high altitudes, the humidity is rather low, meaning that the relative difference can fluctuate a lot, yet very few discrepancies greater than 50 % were found. Nonetheless, there seems to be a negative bias between the model and the lidar at those high altitudes. The higher discrepancies between the lidar and the model during autumn is most likely caused by the use of the older model version as the available data for these months are based on the years 2012-2014. A dry bias of 0.02-0.08 g kg$^{-1}$ has been found between the two model versions, with a higher bias at lower altitudes.

## 6 Conclusions

In this study we employed water vapor profiles from radio soundings, retrievals from AIRS/AMSU instruments on board of Aqua satellite and for the first time the ALADIN/HIRLAM model to calibrate a Raman lidar with water vapor capabilities. The uncertainty of the calibration factor from the radio soundings is <1 % and 2.8 % for the on-site and nearest RS, respectively. The calibration factor derived from the satellite had a discrepancy of 7.4 %, and from the model a 3.9 %. These results lay well within previous studies using the most common calibration technique that of the regression method. If possible, we urge future lidar users to calculate the calibration factor applying a combined regression line to all their cases

simultaneously, since individual cases can have an impact on the derived calibration factor. In our study, this impact was translated to the aforementioned averaged percentages, yet the goal of the lidar calibration is to determine a proper factor as close as possible to the real factor (assuming that of the RS) and this is better obtained with the proposed method.

For the period being studied intercomparisons between the on-site RS and the rest showed that the model and the nearest available soundings are an effective alternative when no soundings are on site, resulting in the smallest deviations after the lidar. We should note here that the model represents the shape of atmospheric WVMR profiles more accurately in the lowest few kilometers than the nearest RS which is located 100 km away from the lidar site. However, care should be taken when using the nearest RS, as non-stable atmospheric conditions between the two sites may lead to inadequate vertical representation of water vapor; for example, when there is sufficient time lag between the RS and the lidar measurement, especially when PBL activity is not constant e.g. between day and night. The highest discrepancies were observed when comparing on-site RS and satellite WVMR. These discrepancies resulted from the larger spatial and poorer vertical resolution of the satellite data. A maximum 6-hour overpass difference from the on-site measurements and the satellite was allowed in this study. This time limit is usually valid for the nearest RS where the 18:00 UTC RS is used, for example, with the 00:00 UTC lidar measurements. As has been shown in Sect. 4.1, the 6-hour delay is acceptable for nighttime comparisons but not when one of the observations is during the day. We also encourage future users to apply the trajectory method when selecting satellite WVMR profiles and not the closest footprint to that of the site of interest. We concluded to better agreement between the RS, the lidar and the satellite when applying this technique.

In addition, 4-year water vapor data from the lidar and the model were adopted to study the water vapor mixing ratio seasonality at the Kuopio site in Eastern Finland. The analysis showed that three humidity states exist in Kuopio: a wet one during summer months where water vapor values of $5.52 \pm 1.01$ g kg$^{-1}$ were observed within the lowest 2 km, a dry state during winter months with concentrations of $1.17 \pm 0.42$ g kg$^{-1}$ and a transition state during spring and autumn with intermediate values. Similar WV vertical structure to that of lidar was found when using modeled WVMR from ALADIN/HIRLAM. In general, the model simulates correctly WVMR for each season although some discrepancies were observed which are attributed partly to a combination of sparser vertical modeled bins and larger averaged area than that of the lidar. There seems to be a negative bias between the model and the lidar at higher altitudes.

## 7 Data availability

The data are available upon request (contact mail: maria.filioglou@fmi.fi).

## Acknowledgments

This project has received funding from Kone foundation and the European Union's Horizon 2020 research and innovation programme under grant agreement No 654109. Elina Giannakaki acknowledges the support of the Academy of Finland

(project no. 270108). We also acknowledge the use of radiosonde data in collaboration with the U.S. Department of Energy as part of the Atmospheric Radiation Measurement (ARM) Climate Research Facility during BAECC (AMF campaign in Hyytiälä, Finland). The authors gratefully acknowledge the support of ACTRIS-TNA during the BAECC campaign held in Hyytiälä and the NOAA Air Resources Laboratory (ARL) for the provision of the HYSPLIT transport and dispersion model

used in this publication. We would also like to thank Eimear M. Dunne for her valuable contribution.

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

**Table 1: Lidar calibration factors and errors derived from RS, satellite and model data during Hyytiälä and Pallas campaign. For satellite overpasses the distance between the lidar site and the selected footprint along with the time difference from the RS launch time is available. Sunrise/sunset times are also shown.**

| Date/Time of RS on site (UTC) | Sunset (UTC) | Sunrise (UTC) | Calibration factors | | | Time difference in satellite overpass (hours) | Satellite selected overpass distance from site (km) |
|---|---|---|---|---|---|---|---|
| | | | RS on site | Model | Satellite | | |
| **Hyytiälä campaign** | | | | | | | |
| 01.04.2014 23:17 | 17:08 | 03:43[+] | 17.43 ± 0.07 | 17.01 ± 0.29 | 19.80 ± 1.93 | 2 | 136.2 |
| 02.04.2014 23:18 | 17:11 | 03:40[+] | 17.14 ± 0.05 | 18.08 ± 0.47 | 18.16 ± 0.80 | 3 | 75.7 |
| 03.04.2014 23:25 | 17:14 | 03:37[+] | 16.71 ± 0.12 | 18.76 ± 0.60 | 15.38 ± 1.22 | 2 | 92.6 |
| 04.04.2014 23:19 | 17:16 | 03:33[+] | 17.14 ± 0.09 | 16.75 ± 0.74 | 28.25 ± 2.82 | 2.5 | 122.5 |
| 08.04.2014 23:22 | 17:27 | 03:20[+] | 16.59 ± 0.08 | 17.95 ± 0.73 | 24.86 ± 1.66 | 2 | 31.4 |
| 09.04.2014 23.20 | 17:30 | 03:17[+] | 16.67 ± 0.10 | 20.95 ± 1.04 | 26.99 ± 1.48 | 1 | 51.1 |
| 15.04.2014 23:30 | 17:46 | 02:58[+] | 17.94 ± 0.09 | 23.58 ± 1.27 | 20.31 ± 0.57 | 2 | 22.4 |
| 22.04.2014 23:22 | 18:05 | 02:36[+] | 17.68 ± 0.13 | 15.61 ± 1.20 | 23.90 ± 1.32 | 2.5 | 158.4 |
| 23.04.2014 23:22 | 18:08 | 02:33[+] | 17.53 ± 0.14 | 21.57 ± 0.92 | 25.91 ± 3.13 | 1.5 | 53.3 |
| 24.04.2014 23:24 | 18:10 | 02:30[+] | 17.64 ± 0.14 | 10.35 ± 0.83 | 19.37 ± 0.96 | 0.5 | 86.2 |
| 25.04.2014 23:21 | 18:13 | 02:27[+] | 17.91 ± 0.11 | 31.76 ± 0.45 | 17.31 ± 0.72 | 1 | 55.6 |
| 26.04.2014 23:22 | 18:16 | 02:24[+] | 17.66 ± 0.12 | 18.94 ± 1.94 | 19.22 ± 0.98 | 2 | 32.0 |
| 27.04.2014 23:18 | 18:19 | 02:21[+] | 17.47 ± 0.09 | 17.38 ± 0.41 | 17.66 ± 0.81 | 1 | 65.9 |
| 29.04.2014 23:21 | 18:24 | 02:14[+] | 17.40 ± 0.08 | 17.51 ± 0.34 | 17.12 ± 0.75 | 1 | 89.2 |
| 01.05.2014 23:27 | 18:30 | 02:09[+] | 17.06 ± 0.11 | 17.14 ± 0.30 | 18.60 ± 0.90 | 2 | 99.1 |
| 02.05.2014 23:22 | 18:33 | 02:06[+] | 17.40 ± 0.16 | 21.46 ± 0.75 | 18.76 ± 0.82 | 1 | 22.2 |
| 12.05.2014 23:25 | 19:00 | 01:37[+] | 17.74 ± 0.07 | 16.31 ± 0.21 | 17.76 ± 0.33 | 1 | 16.9 |
| 14.05.2014 23:21 | 19:06 | 01:32[+] | 17.23 ± 0.11 | 15.98 ± 1.14 | 25.20 ± 1.50 | 1.5 | 65.9 |
| 20.05.2014 23:20 | 19:22 | 01:16[+] | 17.65 ± 0.06 | 17.06 ± 0.10 | 17.19 ± 0.41 | 1 | 89.4 |
| 22.05.2014 23:19 | 19:27 | 01:12[+] | 17.91 ± 0.10 | 18.22 ± 1.78 | 20.23 ± 0.48 | 2.5 | 126.1 |
| 14.06.2014 23:20 | 20:11 | 00:36[+] | 17.51 ± 0.41 | 17.74 ± 0.99 | 18.41 ± 0.90 | 2.5 | 78.6 |
| 04.08.2014 23.26 | 19:01 | 02:01[+] | 17.81 ± 0.15 | 14.55 ± 1.54 | 17.15 ± 1.62 | 1.5 | 40.2 |
| **Pallas campaign** | | | | | | | |
| 02.12.2015 14:31 | 11:25 | 09:03 | 16.96 ± 0.14 | 16.55 ± 0.29 | 12.52 ± 0.65 | 4.5 | 117.6 |
| 02.12.2015 18:49 | 11:25 | 09:03 | 18.85 ± 0.19 | 16.49 ± 0.13 | 19.46 ± 0.05 | 6 | 238.6 |
| 03.12.2015 17:12 | 11:20 | 09:10 | 16.91 ± 0.14 | 16.92 ± 0.30 | - | - | - |
| 04.12.2015 13:54 | 11:14 | 09:16 | 16.93 ± 0.13 | 16.16 ± 1.08 | 18.30 ± 0.83 | 4 | 226.2 |

| 04.12.2015 18:19 | 11:14 | 09:16 | 17.58 ± 0.13 | 17.96 ± 0.51 | 17.73 ± 1.15 | 6 | 224.2 |

[+] Plus one day.

**Table 2: Lidar calibration factors and errors derived from RS, model and satellite data during Kuopio campaign. Calibration factors from the nearest radio sounding site located in Jyväskylä airport (~100 km away) are also shown. For satellite overpasses the distance between the lidar site and the selected footprint along with the time difference from the RS launch time is available. Note: the same satellite overpass has been used on 27[th] of May 2015 where multiple RS launched in one hour interval.**

| Date/Time of RS on site (UTC) | Sunset (UTC) | Sunrise[+] (UTC) | Calibration factors | | | | Time difference in satellite overpass (hours) | Satellite selected overpass distance from site (km) |
|---|---|---|---|---|---|---|---|---|
| | | | RS on site | RS Jyväskylä (18 UTC) | Model | Satellite | | |
| **Kuopio campaign** | | | | | | | | |
| 15.05.2015 22:00 | 19:04 | 01:09 | 17.53 ± 0.13 | 16.47 ± 0.19 | 18.13 ± 0.31 | 17.41 ± 0.57 | 3 | 64.5 |
| 20.05.2015 21:59 | 19:19 | 00:55 | 17.70 ± 0.12 | 18.74 ± 0.41 | 19.32 ± 0.46 | 16.54 ± 0.89 | 2 | 165.0 |
| 21.05.2015 21:58 | 19:22 | 00:52 | 17.99 ± 0.10 | 19.03 ± 0.34 | 17.04 ± 0.45 | 17.38 ± 0.71 | 3 | 114.9 |
| 22.05.2015 22:01 | 19:24 | 00:50 | 17.04 ± 0.22 | 16.08 ± 0.89 | 17.13 ± 0.54 | 19.77 ± 0.69 | 2 | 114.3 |
| 23.05.2015 22:00 | 19:27 | 00:47 | 17.87 ± 0.17 | 16.94 ± 0.35 | 17.44 ± 0.13 | 17.43 ± 0.16 | 2.5 | 164.7 |
| 25.05.2015 22:15 | 19:33 | 00:42 | 17.38 ± 0.13 | 17.18 ± 0.92 | 17.73 ± 0.31 | 17.76 ± 0.93 | 4 | 131.5 |
| 27.05.2015 21:10 | 19:38 | 00:37 | 17.53 ± 0.08 | 17.23 ± 0.34 | 17.49 ± 0.10 | 15.08 ± 0.70 | 3 | 48.4 |
| 27.05.2015 22:08 | 19:38 | 00:37 | 17.58 ± 0.09 | 16.39 ± 0.15 | 16.87 ± 0.29 | 14.15 ± 0.77 | 2 | 48.4 |
| 27.05.2015 23:13 | 19:38 | 00:37 | 17.52 ± 0.07 | 16.43 ±0.17 | 17.27 ± 0.13 | 14.19 ± 0.72 | 1 | 48.4 |
| 28.05.2015 22:03 | 19:41 | 00:34 | 17.42 ± 0.09 | 14.92 ± 0.19 | 16.34 ± 0.59 | 21.42 ± 1.36 | 3 | 80.2 |

[+] Plus one day

**Table 3: Seasonal mean values of WVMR for different atmospheric layers and their standard deviation as calculated from the lidar.**

| Height (km) | Winter (DJF) | | Spring (MAM) | | Summer (JJA) | | Autumn (SON) | |
|---|---|---|---|---|---|---|---|---|
| | Mean (g kg$^{-1}$) | SD (g kg$^{-1}$) | Mean (g kg$^{-1}$) | SD (g kg$^{-1}$) | Mean (g kg$^{-1}$) | SD (g kg$^{-1}$) | Mean (g kg$^{-1}$) | SD (g kg$^{-1}$) |
| 0 − 2 | 1.15 | 0.40 | 2.47 | 0.49 | 5.54 | 1.02 | 2.55 | 0.88 |
| 2 − 4 | 0.41 | 0.06 | 0.91 | 0.23 | 2.34 | 0.71 | 0.97 | 0.22 |
| 4 − 6 | 0.24 | 0.05 | 0.48 | 0.09 | 1.19 | 0.25 | 0.62 | 0.16 |
| 6 − 8 | 0.11 | 0.03 | 0.25 | 0.07 | 0.67 | 0.21 | 0.28 | 0.06 |

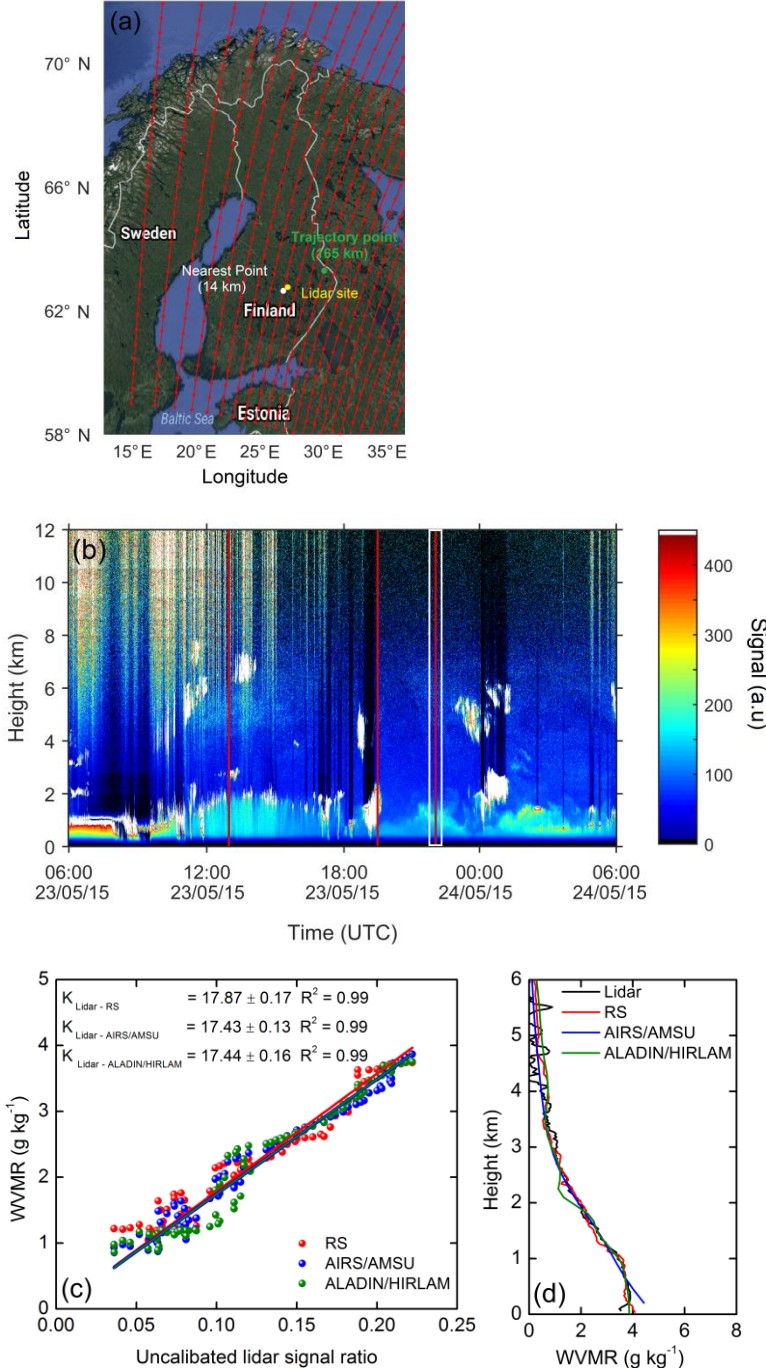

Figure 1: a) Aqua overpass on 23[th] of May 2015. The red dotted lines indicate the flight pattern in the area of interest. The nearest grid point (14 km away from the lidar site) and the selected point (165 km away from the lidar site) at 00:28 UTC are marked with white and green dots, respectively while the lidar site location is marked with a yellow dot. b) Range corrected lidar signals at 1064 nm wavelength between 06:00 UTC on 23[th] and 06:00 UTC on 24[th] of May 2015. The radio soundings performed throughout this period are marked with red lines while the white rectangle shows the period over which the lidar signal is averaged. The night RS at 22:00 UTC is considered for this case. c) Linear regression between the uncalibrated lidar signal ratio and the WVMR from

the RS (red), AIRS-AMSU (blue) and ALADIN/HIRLAM (green). The calibration factor K and the standard error of the slope are reported. The ALADIN/HIRLAM WV profile at 22:00 UTC 23[th] of May 2014 was chosen. d) Respective lidar calibrated water vapor mixing ratios (black), RS (red), AIRS/AMSU (blue) and ALADIN/HIRLAM (green) for the same day.

30

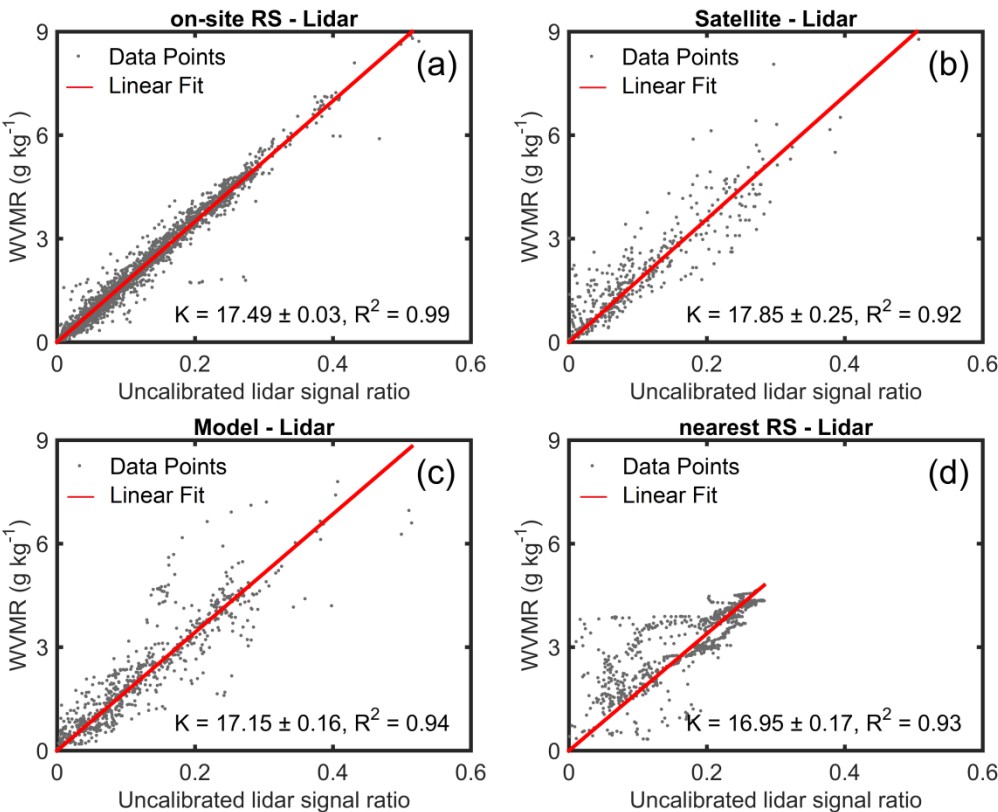

**Figure 2: Overall calculation of the calibration factor including all available cases between the lidar and a) on-site RS b) the satellite, c) the model and d) the nearest RS (for Kuopio site only). Data points are marked as grey dots and the regression as red line. The calibration factor is also shown for each method.**

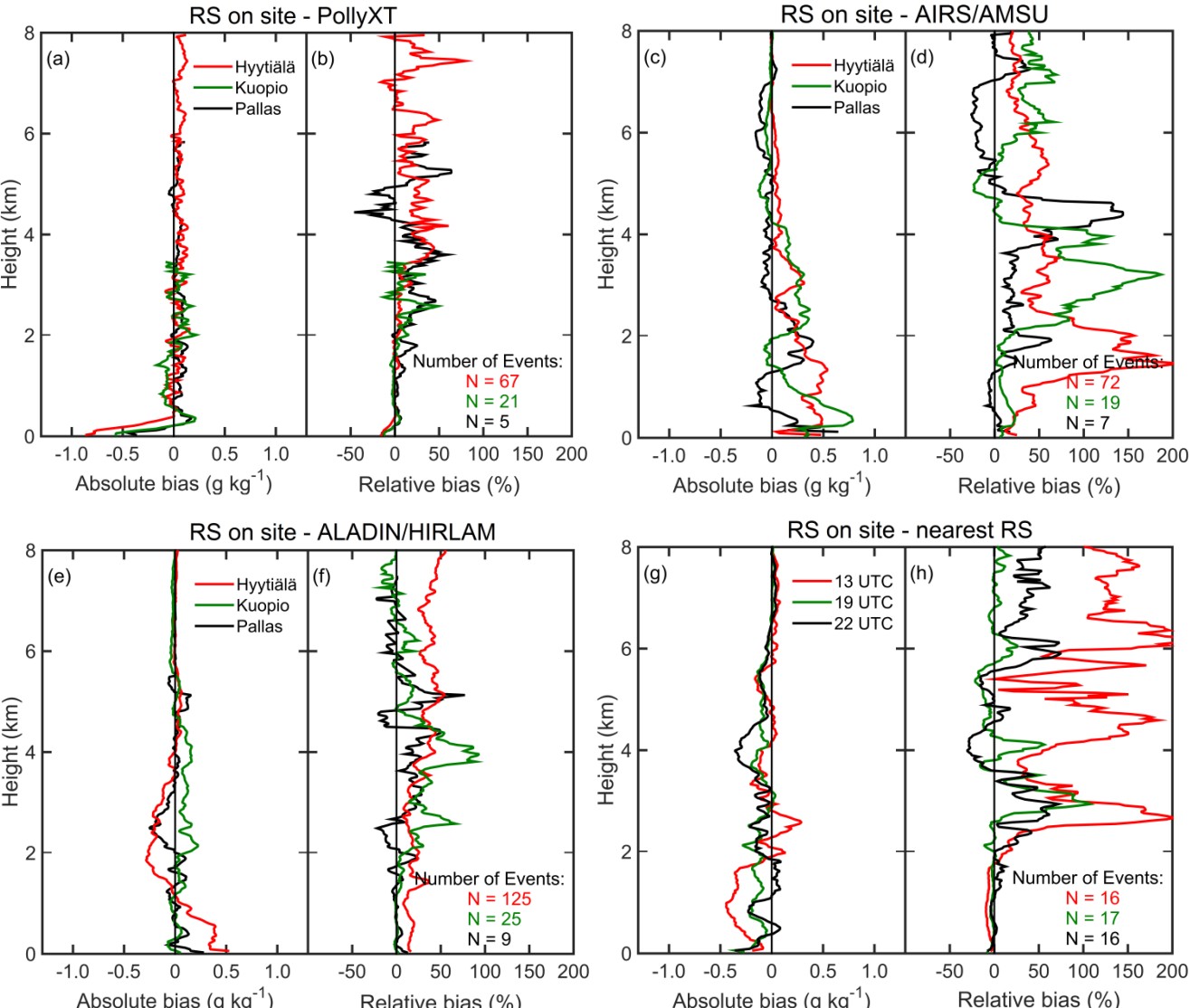

**Figure 3: Comparisons between the on-site radio soundings and Polly$^{XT}$, AIRS/AMSU, ALADIN/HIRLAM and nearest available radio soundings (for Kuopio site only). The various campaigns are indicated with different colors: red for Hyytiälä, green for Kuopio and black for Pallas. For the comparison between on-site RS and the nearest RS (Figures g and h), the different colors indicate the three different on-site RS launch times compared to that of the nearest RS at 18:00 UTC. A 30 min average of the lidar signals is used. The trajectory method is used to select an appropriate WVMR profile from satellite data. The WV profile from ALADIN/HIRLAM model was selected from the nearest time point. Figures a, c, e and g show the mean absolute bias from the on-site RS. Figures b, d, f and h indicate the corresponding mean relative bias. Height is above ground level.**

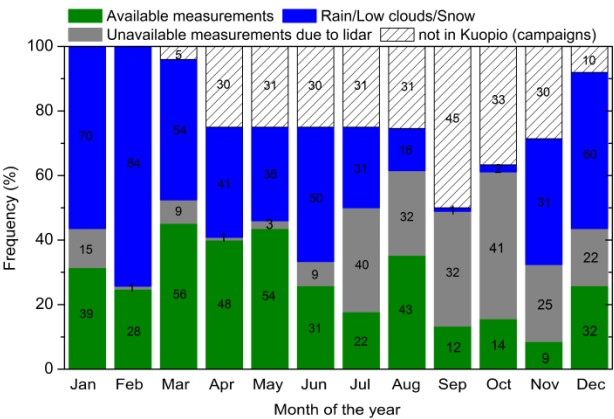

**Figure 4: Monthly percentages of night time profiles for the period between 15[th] of November 2012 and 31[st] of August 2016. The measurements have been categorized depending on their availability as: available, unavailable measurements due to lidar, not in Kuopio (campaigns) and unavailable due to rain/ low clouds/ snow. The numbers on top of the color bars indicate the number of profiles for each category.**

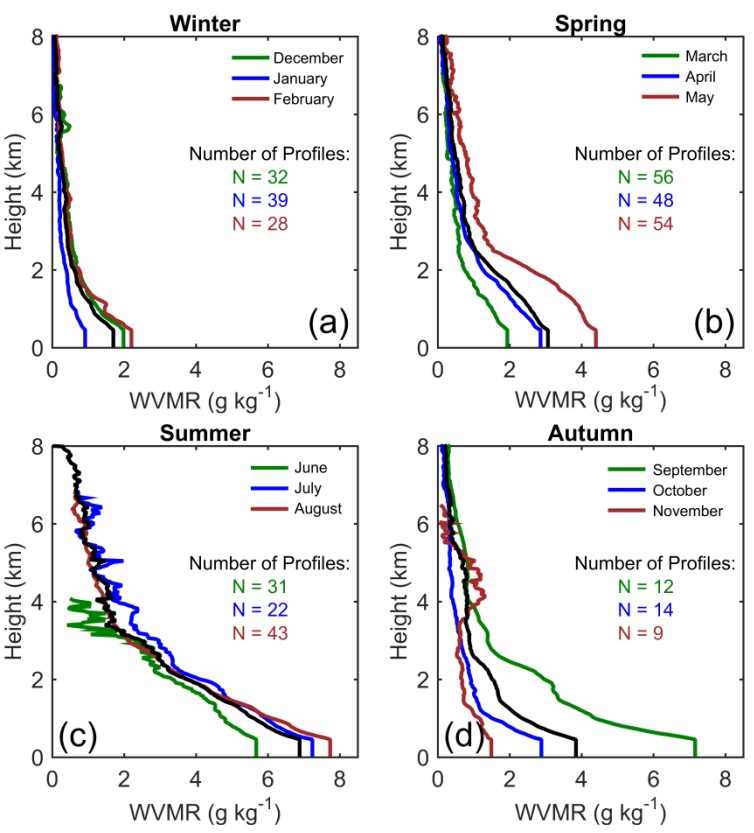

**Figure 5: (a)-(d) Seasonal mean vertical profile of WV mixing ratios (black lines) as seen by the lidar and monthly mean WV mixing ratios (colored lines) for the aforementioned period. Some profiles stop below 8 km due to the SNR limitation.**

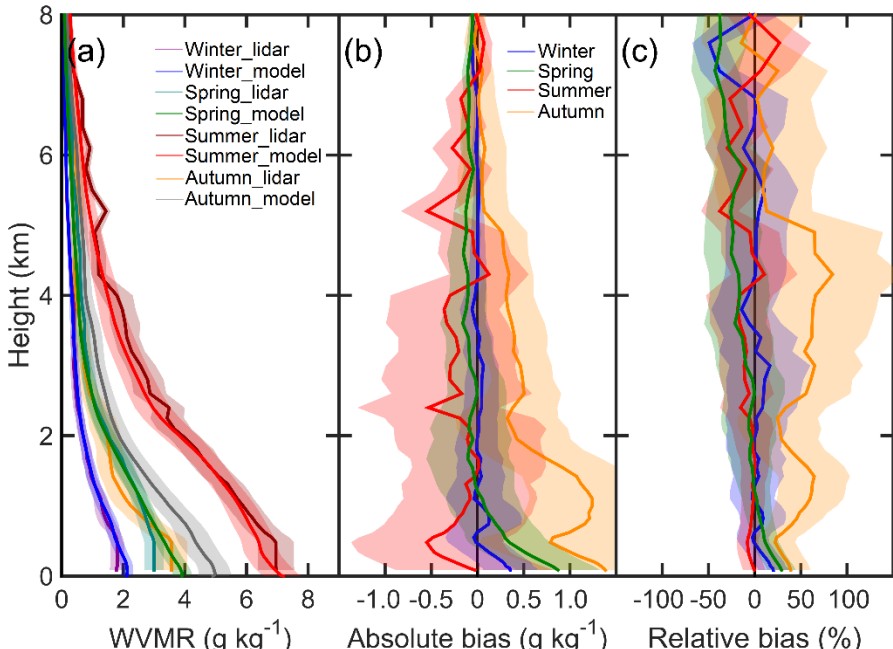

**Figure 6: a) Seasonal water vapor mixing ratios retrieved by the Raman lidar and the model between November 2012 and August 2016. b) Respective seasonal absolute and c) Relative biases. Colored lines show the mean profile and shaded areas the standard deviation.**