# Peer review of "Profiling water vapor mixing ratios in Finland by means of a Raman lidar, a satellite and a model"

_Atmospheric Measurement Techniques, 2017_

## Referee Comment (RC1) · M. Kottas (Referee) · 25 Aug 2017

General Comments

As the article title suggests, the author presents water vapor mixing ratio (WVMR) profiles in Finland, while at the same time, the LiDAR calibration factor for the WVMR is calculated, utilizing three (3) different benchmarks: i) satellite retrievals, ii) radiosondes, and iii) model data. Despite the similar studies that have been conducted in the recent past regarding the calibration factor (e.g. Mamouri, R.E., et al., 2007 | Bhawar, R., et al., 2011), it is important to constantly assess the available tools for optimal LiDAR WVMR retrievals.

[Figure]

Specific Comments

The LiDAR retrievals of WVMR require a priori knowledge of the molecular number density. The source of this knowledge has to be clearly mentioned in the article, since it may have an influence on the final comparison. What is the effect on the LiDAR-derived WVMR, the extraction of temperature/pressure (molecular density) profiles from: i) the AIRS instrument, ii) the radiosonde, and iii) the model?

Technical Corrections

Page 2, Line 1: "therefore" can be omitted. Page 2, Line 10: "to" can be omitted. Page 2, Line 28: "set up" is a verb, while "set-up" or "setup" is the noun. Page 3, Line 14: "where three" → "where the three" Page 7, Line 24: fluctuated Page 7, Line 31: channels

---

## Referee Comment (RC2) · Anonymous Referee #1 · 6 Sep 2017

The paper mostly discusses the calibration of water vapor mixing ratios using Raman lidar measurements with the synergy of radiosonde measurements, satellite retrievals and model simulations. The paper is useful for the evaluation of different options to calibrate the lidar measurements depending on the availability auxiliary information near a lidar station and thus should be considered for publication in AMT. It is well written and structured but the authors should consider the comments below before the acceptance of the manuscript.

General comment:

[Figure]

It missing in the discussion and the conclusions a reference to previous studies that deal with the calibration of lidar WVMRs. Are the values shown applicable to other systems? Are the estimated uncertainties larger or smaller, etc? In general the authors should provide a clear message to other groups that perform lidar measurements of water vapor. The evaluation of different options to calibrate the measurements is such a message but this has to be compared with what is considered common or best practice in the literature. In addition the authors should provide a comment what is the impact of these uncertainties for long-term studies and make a comment how these compare with other sources of water vapor measurements. This would help them to highlight the importance of lidar measurements for long-term studies.

Specific comments:

Page 2, line 1: something is missing from the sentence before "therefore".

Page 4, line 8. Some quantitative information on the uncertainties should be provided here.

Page 4, line 12. Is there any reasoning for this large number of soundings at Hyytiälä compared to other sites? Have these been used in this present study? Did the authors used only the nighttime soundings for the lidar calibration?

Page 4, line 20. Have these data been used only for the campaign period or these are routinely used for the Kuopio measurements?

Page 4, line 22. The authors should be more specific here for the uncertainties, especially for their possible height dependence.

Page 5, line 3. It is not clear here for the reader what would be the impact of these two configurations (if already known) on the water vapor mixing ratios simulated.

Page 5, line 28. Is the assumption to neglect the particle extinction contribution at such heights valid only for the clean conditions prevailing at the certain sites?
Page 6, lines 6 and 10. The authors make use of the terms accuracy and precision in a confusing way. I guess they are referring in both cases to accuracy. Please correct or evaluate more.

Page 6, line 23. The authors should provide here some reasoning why they chose the regression method. Is this a matter of data availability? Do they claim this is better? As written, they give the impression that they made a random choice.

Page 6, line 29. What are the major drifts of the RS at higher altitudes and what is this altitude range? See also a previous relevant comment.

Page 6, lines 30-31. This trajectory-based matching methodology as described is confusing. What do the authors finally use? The closets satellite pixel to the site, or the pixel from which the forward/backward trajectories arrive closest to the site? What distance is considered close? Please clarify and provide a better description.

Page 8, line 2. Something is missing in the sentence after "between the".

Page 8, line 11. Are the larger differences observed below 0.5km affected by the overlap effect on the calibration? How large is this effect and down to what altitude can this be (or has been) corrected?

Page 10, line 1-2. The authors associate the higher discrepancies between the lidar and the model to different model setups. They should provide a comment what can cause this difference. See also a previous comment relevant to the model description.

---

## Author Comment (AC1) · 27 Sep 2017

The authors would like to thank the reviewers for their helpful comments and suggestions that improved our manuscript.

We have included here the manuscript with the proposed changes along with step-by-step answers to the comments. Please note that only major changes have been highlighted (in bold) in the manuscript.

Reviewer: Michael Kottas

Specific Comments

The LiDAR retrievals of WVMR require a priori knowledge of the molecular number density. The source of this knowledge has to be clearly mentioned in the article, since it may have an influence on the final comparison. What is the effect on the LiDAR-derived WVMR, the extraction of temperature/pressure (molecular density) profiles from: i) the AIRS instrument, ii) the radiosonde, and iii) the model?

**The effect that the use of different input data for the molecular density calculation has on the lidar-derived WVMR is minute. The following graph shows the relative difference when calculating the mixing ratios using different molecular densities as from the satellite, the radiosonde, and the model. As reference we have used the lidar WVMR profile calculated with the radiosonde data and then subtracted the rest two. We have included here five of the worst case scenarios found in our dataset in order to see the maximum possible discrepancy, hence someone should expect smaller inconsistencies. We found that <0.1% is the effect between the RS and the satellite calculations of molecular density in the lidar WVMR. The model can introduce slightly bigger discrepancies up to 0.32% but in any case, less than 0.9%. These results are valid for our dataset and refers to nighttime or near-nighttime observations only, valid for the whole atmospheric column up to 8 km.**

[Figure]

**In general, for the lidar WVMR retrievals we have used temperature/pressure information from the radiosonde as this is the most accurate data that we can have. Hence the molecular number density calculations throughout the manuscript was made using the available radiosondes. We clearly state this now**

**in our manuscript along with the expected discrepancy found when using different sources for the molecular density calculation (Section 3.1).**

Technical Corrections

Page 2, Line 1: "therefore" can be omitted.   Page 2, Line 10: "to" can be omitted.

Page 2, Line 28: "set up" is a verb, while "set-up" or "setup" is the noun. Page 3, Line

14: "where three" →"where the three" Page 7, Line 24: fluctuated Page 7, Line 31:

Channels

**We have corrected the manuscript taking into account the technical corrections suggested.**

Reviewer #2

The paper mostly discusses the calibration of water vapor mixing ratios using Raman lidar measurements with the synergy of radiosonde measurements, satellite retrievals and model simulations. The paper is useful for the evaluation of different options to calibrate the lidar measurements depending on the availability auxiliary information near a lidar station and thus should be considered for publication in AMT. It is well written and structured but the authors should consider the comments below before the acceptance of the manuscript.

General comment:

It missing in the discussion and the conclusions a reference to previous studies that deal with the calibration of lidar WVMRs. Are the values shown applicable to other systems? Are the estimated uncertainties larger or smaller, etc? In general the authors should provide a clear message to other groups that perform lidar measurements of water vapor. The evaluation of different options to calibrate the measurements is such a message but this has to be compared with what is considered common or best practice in the literature. In addition the authors should provide a comment what is the impact of these uncertainties for long-term studies and make a comment how these compare with other sources of water vapor measurements. This would help them to highlight the importance of lidar measurements for long-term studies.

**We have addressed all the suggestions mentioned above by adding more discussion to our paper.**

Specific comments:

Page 2, line 1: something is missing from the sentence before "therefore".

**Corrected**

Page 4, line 8. Some quantitative information on the uncertainties should be provided here.

**Added accordingly to reviewer's suggestion**

Page 4, line 12. Is there any reasoning for this large number of soundings at Hyytiälä compared to other sites? Have these been used in this present study? Did the authors used only the nighttime soundings for the lidar calibration?

**The large number of soundings at Hyytiälä site resulted from the longer campaign period, five months compared to one month in Kuopio campaign and 3 months in Pallas campaign along with the more frequent radiosonde launches (4 per day). In the current study as mentioned in the manuscript, only nighttime lidar data have been used hence only nighttime soundings are taken into consideration. For the lidar calibration only cloud-free, nighttime sky is considered.**

Page 4, line 20. Have these data been used only for the campaign period or these are routinely used for the Kuopio measurements?

**The on-site soundings performed during the campaign period in Kuopio were a one-time thing and they are used only for the lidar measurements regarding that period. The operational soundings for Kuopio are the ones performed twice a day at Jyväskylä airport (nearest RS as mentioned in the manuscript). These data are usually available in the lidar measurements.**

Page 4, line 22. The authors should be more specific here for the uncertainties, especially for their possible height dependence.

**Text has been added to the manuscript specifying the valid heights of these biases.**

Page 5, line 3. It is not clear here for the reader what would be the impact of these two configurations (if already known) on the water vapor mixing ratios simulated.

**In Section 5.2, when comparing lidar WVMR with modeled WVMR, we specify the biases found between the two model versions. Regardless of that, we have now included this information when describing the model as well (Section 2.3).**

Page 5, line 28. Is the assumption to neglect the particle extinction contribution at such heights valid only for the clean conditions prevailing at the certain sites?

**According to Whiteman, 2003, the error such assumption can introduce is approximately 2 % every 0.5 of AOD change below 2 km altitude. It has been added in the manuscript.**

Page 6, lines 6 and 10. The authors make use of the terms accuracy and precision in a confusing way. I guess they are referring in both cases to accuracy. Please correct or evaluate more.

**Corrected as suggested.**

Page 6, line 23. The authors should provide here some reasoning why they chose the regression method. Is this a matter of data availability? Do they claim this is better? As written, they give the impression that they made a random choice.

**We have chosen the specific method as it the most accurate technique compared to the rest two which can also make use of most of our measurements. Since, high cloud coverage is valid for all three sites, e.g. mid or high-level clouds, the use of the second method which takes into account the total perceptible water (TPW) from a microwave radiometer wouldn't be applicable, reducing the amount of cases available for the calibration to minimum. We have added the reasoning to the manuscript.**

Page 6, line 29. What are the major drifts of the RS at higher altitudes and what is this altitude range? See also a previous relevant comment.

**The major drifts of the RS refers to the device being carried away by the wind speed and its direction. This implies that the lidar and the RS end up measuring different atmospheric layers which have an effect in the calculated slope hence the calibration factor. Figure 3b shows the difference in the WVMR as measured by the RS and the lidar. Better agreement is found in the lowermost altitudes up to 4 km reaching up to 35 % above that range. One of the reasons can be the drift of the device under unstable atmospheric conditions. Of course, the mixing ratios at such heights is rather low and this acts synergistically. A second reason for this calibration upper height limitation is the SNR values of the lidar signal. We have set the limit in order to be well within those boundaries for most of our cases.**

Page 6, lines 30-31. This trajectory-based matching methodology as described is confusing. What do the authors finally use? The closest satellite pixel to the site, or the pixel from which the forward/backward trajectories arrive closest to the site? What distance is considered close? Please clarify and provide a better description.

**We use the satellite profile as pointed by the trajectory analysis. We do not use the closest to the site satellite profile unless it is indicated by the trajectories. In our study we have allowed any distance as long as there was a satellite point to the trajectory. This can be one reason for having calculated calibration factors that do not much the RS yet, we have found that this method describes more accurately the WVMR of the sites than the closest satellite profile. On the applicability of this method to all sites, we should note here that a distance can be considered close or not depending on the surroundings. In our case, Pallas site is a hilly region which can easily modify the mixing ratios, at least, in the lowermost part although we still found that the 200 km away satellite profile matches better with the RS and the lidar. We have provided a better description in the manuscript.**

Page 8, line 2. Something is missing in the sentence after "between the".

**Corrected**

Page 8, line 11. Are the larger differences observed below 0.5km affected by the overlap effect on the calibration? How large is this effect and down to what altitude can this be (or has been) corrected?

**As described in Section 4, for the calibration we have considered lidar signals above 0.5 km. Although the two signals used for the water vapor technique should have the same overlap, hence the signal can be ideally used down to the first bin, our measurements show inconsistencies with the RS and for that reason we excluded the first 500 m from the calibration. In the seasonal analysis we assumed that the first 500 m are well-mixed, hence the value at that point was used down to the surface. The effect in our case was less than 17 % (see Fig 3b). Theoretically, it can be corrected down to surface, practically, it needs continuously available measurements from a reference instrument to study whether this pattern is systematic and stable with time. Such work is out of the scope of this paper but we are aware of it.**

Page 10, line 1-2. The authors associate the higher discrepancies between the lidar and the model to different model setups. They should provide a comment what can cause this difference. See also a previous comment relevant to the model description.

We have associated the different behavior during autumn to be mainly caused by the older model version used as our dataset for autumn includes only cases from that model period compared to the rest seasons. We should also consider that these are three different locations with different and relatively short time periods, which each can have different bias characteristics even in the same model version. Regardless of that, the main change in the new version is related to cloud microphysics especially in cold conditions. In such conditions, the clouds, in the old version, were mainly in ice phase, which turned too rapidly into falling snow. This resulted that actual clouds "came down" as snow too rapidly removing moisture from atmosphere. Therefore, the reduction of dry bias could be side effect of that process (see also section 2.3)

[revised manuscript text omitted]